# Highly Red Light-Emitting Erbium- and Lutetium-Doped Core-Shell Upconverting Nanoparticles Surface-Modified with PEG-Folic Acid/TCPP for Suppressing Cervical Cancer HeLa Cells

**DOI:** 10.3390/pharmaceutics12111102

**Published:** 2020-11-17

**Authors:** Kyungseop Lim, Hwang Kyung Kim, Xuan Thien Le, Nguyen Thi Nguyen, Eun Seong Lee, Kyung Taek Oh, Han-Gon Choi, Yu Seok Youn

**Affiliations:** 1School of Pharmacy, Sungkyunkwan University, 2066 Seobu-ro, Jangan-gu, Suwon, Gyeonggi-do 16419, Korea; imkyung424@naver.com (K.L.); hkkim0319@naver.com (H.K.K.); lexuanthien.dkh@gmail.com (X.T.L.); ngtnguyen1710@gmail.com (N.T.N.); 2Division of Biotechnology, The Catholic University of Korea, 43 Jibong-ro, Bucheon, Gyeonggi-do 14662, Korea; eslee@catholic.ac.kr; 3College of Pharmacy, Chung-Ang University, 84 Heukseok-ro, Dongjak-gu, Seoul 06974, Korea; kyungoh@cau.ac.kr; 4College of Pharmacy, Hanyang University, 55 Hanyangdaehak-ro, Sangnok-gu, Ansan 15588, Korea; hangon@hanyang.ac.kr

**Keywords:** upconverting nanoparticles, photodynamic therapy, tetrakis(4-carboxy-phenyl)porphyrin, near infrared, cancer, hypoxia

## Abstract

Photodynamic therapy (PDT) combined with upconverting nanoparticles (UCNPs) are viewed together as an effective method of ablating tumors. After absorbing highly tissue-penetrating near-infrared (NIR) light, UCNPs emit a shorter wavelength light (~660 nm) suitable for PDT. In this study, we designed and prepared highly red fluorescence-emitting silica-coated core-shell upconverting nanoparticles modified with polyethylene glycol (PEG_5k_)-folic acid and tetrakis(4-carboxyphenyl)porphyrin (TCPP) (UCNPs@SiO_2_-NH_2_@FA/PEG/TCPP) as an efficient photodynamic agent for killing tumor cells. The UCNPs consisted of two simple lanthanides, erbium and lutetium, as the core and shell, respectively. The unique core-shell combination enabled the UCNPs to emit red light without green light. TCPP, folic acid, and PEG were conjugated to the outer silica layer of UCNPs as a photosensitizing agent, a ligand for tumor attachment, and a dispersing stabilizer, respectively. The prepared UCNPs of ~50 nm diameter and −34.5 mV surface potential absorbed 808 nm light and emitted ~660 nm red light. Most notably, these UCNPs were physically well dispersed and stable in the aqueous phase due to PEG attachment and were able to generate singlet oxygen (^1^O_2_) with a high efficacy. The HeLa cells were treated with each UCNP sample (0, 1, 5, 10, 20, 30 μg/mL as a free TCPP). The results showed that the combination of UCNPs@SiO_2_-NH_2_@FA/PEG/TCPP and the 808 nm laser was significantly cytotoxic to HeLa cells, almost to the same degree as naïve TCPP plus the 660 nm laser based on MTT and Live/Dead assays. Furthermore, the UCNPs@SiO_2_-NH_2_@FA/PEG/TCPP was well internalized into HeLa cells and three-dimensional HeLa spheroids, presumably due to the surface folic acid and small size in conjunction with endocytosis and the nonspecific uptake. We believe that our UCNPs@SiO_2_-NH_2_@FA/PEG/TCPP will serve as a new platform for highly efficient and deep-penetrating photodynamic agents suitable for various tumor treatments.

## 1. Introduction

Photodynamic therapy (PDT) has attracted great interest as an effective means of treating cancers [1,2,3]. PDT utilizes photosensitizers (PS) that absorb visible light and convert molecular oxygen to reactive oxygen species (ROS) such as singlet oxygen (^1^O_2_) or other free radicals [2,4]. The generated ROS oxidizes cellular proteins, leading to tumor ablation via such mechanisms as direct tumor cell killing, damage to tumor vasculature, and tumor suppression [5,6]. Cancer cells can be killed by the photodynamic process through several mechanisms of apoptosis, necrosis, or autophagy [5]. Necrosis is triggered by a high dose of PS/light and involves cytoplasm swelling, organelle devastation, or membrane disruption, whereas apoptosis is characterized by cell shrinkage, chromosomal DNA fragmentation, or chromatin condensation [5,7].

In the context of an intratumoral injection or tumor targeting approach, PDT has the clear advantage of localizing tumor therapy by irradiating light selectively on the tumor location, with less invasiveness and fewer side effects [5]. In addition, the treatment is of short duration, is inexpensive, and induces an immune response that can produce synergistic effects on anticancer activity [8,9]. However, the clinical use of PDT is often restricted because most photosensitizers are activated around a wavelength of ~660 nm, which cannot penetrate the skin deeply when compared to near-infrared (NIR) light (~780–940 nm) [1,10,11,12].

Upconverting nanoparticles (UCNPs) have unique optical properties that convert NIR light to UV or visible light [13], which enables many biomedical applications, such as bioimaging, photodynamic therapy, and combined cancer therapy [14,15,16,17]. The concept of UCNPs is based on the anti-Stokes phenomenon, in which two or more photons are absorbed and a single quantum of energy is emitted, generating short wavelength light from long wavelength light [18,19,20]. The structure of UCNPs is a matrix of alkali metal compounds in which the respective lanthanide elements are embedded [21,22]. In UCNPs, lanthanide elements consist of a sensitizer that absorbs long wavelengths of light and an activator that emits short wavelengths of light. In particular, the core@shell structure that separates the activator and the sensitizer is widely used to maximize the fluorescence efficiency of UCNPs [23,24,25,26]. The clinical significance of UCNPs requires the strong permeability of NIR light because the absorption of biological window I (700–980 nm) light is low in the skin and body [27]. Consequently, UCNPs have been utilized as a light-generating source (~660 nm) subsequent to their absorption of well-permeating NIR light (>800 nm), which overcomes the limitation of triggering PS in the body [28,29,30,31].

Herein, we sought to design and prepare highly red fluorescence light-emitting core-shell upconverting nanoparticles surface-modified with silica, followed by PEG-folic acid (FA/PEG) and TCPP (UCNPs@SiO_2_-NH_2_@FA/PEG/TCPP), as an efficient photodynamic agent for killing tumor cells. The UCNPs@SiO_2_-NH_2_@FA/PEG/TCPP was fabricated in a core-shell structure by doping with erbium/lutetium to mainly emit ~660 nm to activate the adjoining TCPP, and the folic acid ligand was conjugated as a targeting moiety of HeLa cells. Analytical/morphological investigations to identify the UCNPs@SiO_2_-NH_2_@FA/PEG/TCPP in vitro cytotoxicity were conducted to evaluate the antitumor potential, and an evaluation of the UCNP penetration in HeLa cell spheroids was conducted to achieve this aim (Figure 1).

## 2. Materials and Methods

### 2.1. Materials

Erbium(III) acetate hydrate, lutetium(III) acetate hydrate, yttrium(III) acetate hydrate, ytterbium(III) acetate hydrate, neodymium(III) acetate hydrate, ammonium fluoride (NH_4_F), oleic acid (90%), 1-octadecene (90%), IGEPAL^®^ CO-520, and ammonium hydroxide solution (28–30%) were purchased from Sigma–Aldrich (St. Louis, MO, USA). Tetraethyl orthosilicate (TEOS), 3-aminopropyltriethoxy-silane (APTES), and tetrakis(4-carboxyphenyl)porphyrin (TCPP) were purchased from Tokyo Chemical Industry (TCI) Co., Ltd. (Toshima, Kita-ku, Tokyo, Japan). mPEG_2k_-SC and FA/PEG_5k_-NHS were purchased from Biochempeg (Watertown, MA, USA). The HeLa (ATCC^®^ CCL-2™) human adenocarcinoma cell line was obtained from the American Type Culture Collection (Rockville, MD, USA). DMEM and fetal bovine serum (FBS) were purchased from Capricorn (Ebsdorfergrund, Hesse, Germany). Trypsin-EDTA and penicillin-streptomycin (P/S) solution were purchased from Corning (Corning, NY, USA). The LIVE/DEAD^TM^ viability/cytotoxicity assay kit, LysoTracker^TM^ Green DND-26, and singlet oxygen sensor green reagent (SOSG) were purchased from Thermo Fisher Scientific (Waltham, MA, USA). All other reagents were obtained from Sigma–Aldrich unless otherwise indicated.

### 2.2. Synthesis of Core UCNPs for Red Emission

Upconverting nanoparticles of ~30 nm were manufactured by slight modification of an existing procedure [32,33,34,35,36,37]. Briefly, 3 mmol of Er(CH_3_CO_2_)_3_ in 5 mL of methanol was added to a round bottom flask containing 12 mL of oleic acid (OA) and 45 mL of octadecene (ODE). Methanol was evaporated at 110 °C under vacuum for 20 min, and the reaction temperature was maintained at 150 °C for 1 h under argon flow to obtain a clear solution. After cooling, methanol solution containing 12 mmol NH_4_F and 7.5 mmol NaOH was added slowly to the reaction vessel and kept at 50 °C for 30 min. After removing methanol using the same evaporation process, the vessel was heated to 310 °C at the rate of 10–15 °C/min and maintained at this temperature for 90 min under an argon atmosphere. The mixture was cooled to room temperature, and an equal volume of acetone was added to precipitate the product UCNPs. The resulting suspension was centrifuged at 6554 g for 10 min. The pellet was collected and redispersed in 20 mL of n-hexane. This mixture was centrifuged at 1000 g for 5 min to remove large particles. Finally, the supernatant containing fine UCNPs was harvested and placed in storage at 4–8 °C for further experiments.

### 2.3. Synthesis of Core-Shell UCNPs for Red Emission

The core-shell upconverting nanoparticles were prepared by slight modification of a procedure described elsewhere [37]. Briefly, 1 mmol of Lu(CH_3_CO_2_)_3_ in 2 mL of methanol was added to the round bottom flask containing 8 mL of OA and 30 mL of ODE. Methanol was evaporated at 110 °C under vacuum for 10 min, and the temperature was maintained at 150 °C for 1 h under argon flow to obtain a clear solution. After cooling, 225 mg of the as-synthesized core UCNPs in n-hexane was added to the solution and heated to 110 °C for 30 min under vacuum to remove n-hexane. The temperature was decreased to 50 °C prior to the slow addition of 4 mmol NH_4_F and 2.5 mmol NaOH in methanol. After a 30 min incubation period at 50 °C, methanol was evaporated using the process described above. Next, the reaction vessel was heated to 310 °C at a rate of 10–15 °C /min and maintained at this temperature for 90 min under an argon atmosphere. The mixture was cooled to room temperature, and an equal volume of acetone was added to precipitate UCNPs. The resulting suspension was centrifuged at 6554 g for 10 min. The pellet was collected and redispersed in 20 mL of hexane. This mixture was centrifuged at 1000 g for 5 min to remove large particles. Finally, the supernatant containing fine UCNPs was harvested and placed in storage at 4–8 °C for further studies.

### 2.4. Synthesis of Core@shell Upconverting Nanoparticles (UCNPs) for Green Emission

For the UCNP core structure, 3 mmol each of Y(CH_3_CO_2_)_3_, Yb(CH_3_CO_2_)_3_, Er(CH_3_CO_2_)_3_, and Nd(CH_3_CO_2_)_3_ [by molar ratio, 68.5% Y, 30% Yb, 0.5% Er, 1% Nd] was dissolved in 5 mL methanol. The core synthesis process is the same as that described above. All remaining steps were performed according to the same procedure used for the synthesis of the regular core UCNP structure described above. For the UCNP shell structure, 1 mmol each of Y(CH_3_CO_2_)_3_ and Nd(CH_3_CO_2_)_3_ [by molar ratio, 80% Y, 20% Nd] was dissolved in 2 mL methanol. All remaining steps are the same as those used for the synthesis of the regular shell UCNP structure described above [35,38].

### 2.5. Surface Modification of UCNPs

Amine-functionalized UCNPs (UCNPs@SiO_2_-NH_2_) were prepared through a series of steps described elsewhere [39]. A portion (25 mL) of UCNP solution (2 mg/mL in hexane) was mixed with 1200 mg of IGEPAL^®^ CO-520 in an ultrasound bath. A 200 μL aliquot of ammonium hydroxide solution (30%) was added to the mixture and gently shaken. The reaction started by the addition of TEOS (150 μL, 0.7 mmol) under stirring and continued overnight at room temperature. The resulting UCNPs@SiO_2_ was purified by centrifugation with ethanol (3 × 8500 rpm, 10 min). Amino groups on UCNPs@SiO_2_ were functionalized with APTES (750 μL, 3.4 mmol) in 25 mL of ethanol. This mixture was stirred overnight at room temperature. The UCNPs@SiO_2_-NH_2_ nanoparticle fraction was centrifuged (3 × 8500 rpm, 10 min), harvested, and redispersed in 25 mL distilled water (DW).

### 2.6. Preparation of UCNPs@SiO_2_-NH_2_@FA/PEG/TCPP

UCNPs@SiO_2_-NH_2_ was conjugated with TCPP, mPEG_2k_-SC-NHS, and FA/PEG_5k_-NHS using a slight modification of the procedure described elsewhere [40,41]. To synthesize TCPP-NHS, TCPP (0.2 mmol), N-dicyclohexylcarbodiimide (DCC, 0.3 mmol), *N*-hydroxysuccinimide (NHS, 0.3 mmol), and triethylamine (TEA, 0.3 mmol) were mixed in 5.3 mL dimethyl sulfoxide (DMSO), and the reaction was allowed to continue in the dark for 24 h [2,42]. After removing the precipitate of DCU, the product was stored at −80 °C. UCNPs@SiO_2_-NH_2_ (10 mg), FA/PEG_5k_-NHS (0.1 mg), and TCPP-NHS (1 mg) were mixed in PBS (pH 8.0). After 1 h, mPEG_2k_-SC (2 mg) was added to the resulting solution and stirred for 12 h. The mixture was centrifuged (8500 rpm, 20 min) and redispersed in DW.

### 2.7. Characterization of Various UCNPs

The particle size of a series of UCNPs was determined by dynamic light scattering (DLS) (Zetasizer Nano-S90, Malvern Instruments Ltd., Worcestershire, UK) at a 90° scattering angle. Irradiation by an 808 nm laser (Laserlab CO, Gyeonggi-do, Korea) was used to identify the fluorescence emission of UCNPs. The morphology of UCNPs was examined using an ultrahigh-resolution analytical electron microscope (HR-TEM) (JEM-3010, JEOL Ltd., Tokyo, Japan). A multimode microplate reader (Synergy^TM^ Neo2, BioTek Instruments, Winooski, VT, USA) was used for the UV-Vis-NIR analysis of the core and shell structures.

### 2.8. Singlet Oxygen Generation of UCNPs@SiO_2_-NH_2_@FA/PEG/TCPP

Singlet oxygen sensor green (SOSG) reagents were used to determine the singlet oxygen generation from UCNPs@SiO_2_-NH_2_@FA/PEG/TCPP [2,43]. Aliquots (2 mL) of samples of DW, free TCPP, UCNPs@SiO_2_-NH_2_@FA/PEG, and UCNPs@SiO_2_-NH_2_@FA/PEG/TCPP were mixed with the SOSG reagents (final concentration: 2.5 μM) in a 24-well plate. An 808 nm laser (1.5 W/cm^2^) irradiated the respective sample wells for DW, UCNPs@SiO_2_-NH_2_@FA/PEG, and UCNPs@SiO_2_-NH_2_@FA/PEG/TCPP, and a 660 nm laser (10 mW/cm^2^) was used for free TCPP. While irradiating the laser for 60 min, a 100 μL aliquot of each sample was collected every 10 min, and the fluorescence intensities were determined using a multimode microplate reader (excitation: 485 nm, emission: 538 nm).

### 2.9. Cytotoxicity Evaluation of UCNPs@SiO_2_-NH_2_@FA/PEG/TCPP

The cytotoxicity of a series of UCNPs was evaluated by an MTT and a Live/Dead viability assay kit in HeLa cells using a slightly modified previously described procedure [44,45,46]. For the MTT assay, 100 μL of HeLa cancer cells (1 × 10^5^ cells/mL) was seeded into the wells of a 96-well plate (SPL Life Sciences CO., LTD) and incubated for 24 h. The HeLa cells were treated with each UCNP sample (0, 1, 5, 10, 20, 30 μg/mL as a free TCPP) in DMEM containing 1% FBS and 1% P/S. The concentration of conjugated TCPP onto UCNPs was determined at 660 nm using a UV-Vis spectrometer (VersaMax^TM^ microplate reader, Molecular Devices, CA, USA) because the TCPP absorbance (660 nm) was not overlapped with those of UCNPs alone and folic acid. After a 4-h incubation, cells were rinsed twice and replenished with fresh DMEM. Hypoxic conditions were obtained by incubating in a hypoxic chamber (5% CO_2_, 95% N_2_) for 4 h. Each sample was irradiated with a 660 nm laser (10 mW/cm^2^) or 808 nm laser (1.5 W/cm^2^) for 30 min and then incubated for 24 h. After a treatment with 0.5 mg/mL of MTT reagents for 2 h, MTT reagents were removed, and 100 μL of dimethyl sulfoxide (DMSO) was added for 20 min. Finally, the absorbance intensity at 562 nm was analyzed using a UV-Vis spectrometer.

Live/Dead^TM^ viability/cytotoxicity kits were used to visualize the death and viability of HeLa cells. For this, 100 μL of HeLa cancer cells (1 × 10^5^ cells/mL) was seeded in a 12-well plate (Ibidi, Lochhamer, Germany) and further incubated for 24 h. The subsequent steps were performed according to the same protocol as for the MTT assay. At 24 h after laser irradiation, cells were washed with PBS twice and stained with Live/Dead assay solution (4 mL of PBS, 8 μL of ethidium homodimer-1, 2 μL of calcein-AM) for 30 min. Stained cells were observed using a confocal laser scanning microscope (CLSM) (LSM510, Carl Zeiss Meditec AG, Jena, Germany).

### 2.10. Cellular Uptake of UCNPs in HeLa Cells and Spheroids

To investigate the cellular uptake of UCNPs@SiO_2_-NH_2_@FA/PEG/TCPP, 150 μL of HeLa cells (1 × 10^5^ cells/mL) was seeded in an 8-well plate (Ibidi, Lochhamer, Germany) and incubated for 24 h [46]. HeLa cells were treated with each UCNP sample (20 μg/mL as a free TCPP). At predetermined time points (10, 20, 30, and 60 min), cells were washed three times with PBS and stained with LysoTracker^TM^ Green and DAPI [47,48]. The cells were observed using a CLSM (excitation of TCPP: 633 nm, emission of TCPP: 650–680 nm) [49]. Separately, the internalization of UCNPs@SiO_2_-NH_2_@FA/PEG/TCPP was evaluated in three-dimensional HeLa cell spheroids [2,45,46]. Aliquots (100 μL) of HeLa cells (3 × 10^4^ cells/mL) in DMEM (containing 1% FBS, 1% P/S) were seeded in V-shaped 96-well plates designed to induce the formation of 3D multicellular spheroids (Shimadzu Scientific Instruments, Kyoto, Japan) and incubated for 36 h. The cell spheroids were treated with UCNPs@SiO_2_-NH_2_@FA/PEG/TCPP (30 μg/mL of TCPP). After a 24-h incubation, the spheroids were transferred to an 8-well plate with PBS, and cells were observed using a CLSM.

### 2.11. Data Analysis

Data are presented as the mean ± standard deviation (SD). The significance of differences was determined by a Student’s *t*-test. *p*-values < 0.05 were considered statistically significant.

## 3. Results and Discussion

### 3.1. Synthesis and Characterization of Core@shell UCNPs

The conventional UCNPs doped with the core or shell element of yttrium (Y), ytterbium (Yb), and neodymium (Nd) have split fluorescence lights of ~545 nm and 660 nm, which is considered to be a main reason for a reduced PDT efficiency [37]. To this end, the first specific aim of this study was to develop a prototype of UCNPs@SiO_2_ that emitted ~660 nm red light and was surface-modified with folic acid for an efficient PDT treatment. As a first step, we aimed to produce a hexagonal core of UCNPs to achieve optimal fluorescence efficacy values [50]. As shown in Figure 2A, changing the molar ratio of OA and ODE resulted in a significant morphological change of core UCNPs: the core UCNPs were spherical at a ratio of 3:7 but had sharpened edges and exhibited an optimal hexagonal shape at a ratio of 4:15. Therefore, this ratio was chosen for the core UCNPs.

The surface characteristics of UCNPs are of prime importance because lanthanide-doped core-only UCNPs have a relatively weak upconverting efficiency. Among several methods, the utilization of a core/shell structure is considered the most effective for enhancing the efficiency [14]. The shells of UCNPs were made of lutetium using Lu(CH_3_CO_2_)_3_ to produce UCNPs with a red fluorescence. At molar ratios (core to shell = Er(CH_3_CO_2_)_3_ to Lu(CH_3_CO_2_)_3_) of 1:0.5, 1:1, and 1:1.5, the morphologies of core@shell UCNPs were observed by TEM. Increasing the lutetium amount led to a size increase of core@shell UCNPs. However, the empty shell debris not attached to the surface of core@shell UCNPs was shown at ratios >1:1, presumably due to excess lutetium shell material. This debris resulted in an inappropriate overall shape for further preparation. Based on the core@shell structure and size distribution, UCNPs were most uniform at a core to shell ratio of 1:0.5 (core:shell) (Figure 2B). The results of HR-TEM and dynamic light scattering (DLS) showed that the particle sizes for the core and core@shell of UCNPs were 28.4 ± 6.0 nm and 33.6 ± 7.7 nm, respectively (Figure 3A,B). The size increase at each step of the core, shell, and silica layer indicated that our UCNPs were well-doped and well-coated with Er, Lu, and SiO_2_ and that this unique structure would play a critical role in photodynamic therapy and tumor recognition.

### 3.2. Surface Modification of Core@shell UCNPs

The physical stability of UCNPs was a critical consideration in targeting ligand conjugation and further drug delivery because lanthanide-doped and oleate-capped UCNPs are hydrophobic and weakly water-dispersible [51,52]. Therefore, the hydrophilic surface modification of such UCNPs is essential for subsequent pharmaceutical stability and biomedical applications [52]. Many methods have been utilized to achieve a hydrophilic UCNP surface by NOBF_4_-(nitrosonium tetrafluoroborate)-based two-step exchange using various ligands, such as bisphosphonoglycine, citrate, polyacrylic acid (PAA), polyethylene glycol, and polyallylamine (PAAm) [51]. In this study, silica coating is considered the most useful method to address the surface hydrophobicity of UCNPs and to provide an advantage for further conjugation steps. The silica layer converts hydrophobic UCNPs to hydrophilic/dispersible UCNPs in aqueous media. It is also easily modified with various chemical groups (e.g., −COOH, −NH_2_, −SH) that provide conjugation with functional agents or biological molecules [39,53,54]. To introduce a silica layer on the UCNPs, the well-established reverse micelle method was used: SiO_2_ grew on the surface of Igepal-stabilized UCNPs via the ammonia-catalyzed polymerization of TEOS [54]. The thickness of the silica coating layer was about 10 nm, showing a total size of 53.6 ± 16.7 nm for core@shell-UCNP@SiO_2_, an increase of 20 nm over the prior step (Figure 3C). Additionally, the resulting size of core@shell-UCNP@SiO_2_ appeared to have the potential for tumor accumulation because nanoparticles of 50–100 nm have a high tumor targetability [55,56].

UCNPs absorb light in the infrared region and emit light in various visible regions, which enables their application to PDT [40,41], PTT [57,58], and bioimaging [17,59,60]. For PDT using porphyrin ring structured PS in particular, it is of prime importance that UCNPs absorb long wavelength light (~800 nm) and maximize the fluorescence efficiency of red light (600–700 nm), since most PS are activated near 660 nm in the red region. Conventional UCNPs made use of lanthanide elements (Y, Yb, Er, and Nd) that showed a strong emission of green light (Figure 4A). Green fluorescence is more applicable to bioimaging than to PDT. In contrast, our simple UCNPs made of the Er core and Lu shell displayed a strong red fluorescence light emission (Figure 4B). Furthermore, this red light was maintained significantly despite the 10 nm silica coating layer. In agreement with a previous result by He et al. [37], this ability of the core@shell UCNPs might enable the unambiguous strengthening of the PDT effect of TCPP. Likewise, core@shell UCNPs showed peak absorption at around 800 nm NIR light, which was much greater than that of core-only UCNPs, indicating the importance of the lutetium shell (Figure 5A). The result obviously suggested that these UCNPs would have a great effectiveness in generating the relevant light (~660 nm) that is useful for further in vivo PDT, accompanied by the irradiation of the deep-penetrating 808 nm NIR laser.

### 3.3. Preparation and Characterization of UCNPs@SiO_2_@FA/PEG/TCPP

The surface hydroxyl group (−OH) in UCNPs@SiO_2_ was replaced with an amino group (−NH_2_) by adding APTES, which was a preliminary step to conjugation with TCPP-NHS, PEG_2k_-NHS, and FA/PEG_5k_-NHS. A 5k length (molecular weight-based) of folic acid-PEG was chosen as it was longer than naïve PEG (2k) (present solely as a dispersing stabilizer) to locate at the outermost surface of UCNPs. To overcome these problems, we developed UCNPs@SiO_2_-NH_2_@FA/PEG/TCPP conjugates. The use of UCNPs solved the problem of skin permeability, and PEG could overcome the poor solubility of TCPP [29,30,61]. In addition, the targetability of nanoparticles was increased by the conjugation of folic acid [62]. Folic acid is a member of the B vitamin group, and various cancer cells, such as those of ovarian and cervical cancer, express folic acid receptors [63,64]. When folic acid is conjugated to nanoparticles, the endocytosis process can increase the rate of absorption into the tumor [62].

As shown in Figure 5B, amine group introduction in UCNPs@SiO_2_ resulted in a zeta potential change from −29.8 to +10.1 mV. However, the UCNPs@SiO_2_-NH_2_ tended to aggregate due to its relatively weak charges, and naïve PEG and FA/PEG were quickly conjugated to UCNPs@SiO_2_-NH_2_. The zeta potentials of UCNPs@SiO_2_@FA/PEG and UCNPs@SiO_2_@FA/PEG/TCPP were −14.3 and −34.5 mV, respectively (Figure 5B). As shown in Figure 4C, the increase in the feed amount of naïve PEG_2k_ (0.1–2.0 mg to 10 mg UCNPs@SiO_2_) showed a clearly improved dispersibility of the relevant UCNPs@SiO_2_: UCNPs@SiO_2_ precipitated in aqueous media at a PEG_2k_ amount of 0.1–0.2 mg, whereas UCNPs@SiO_2_ was very stable and dispersible at a PEG_2k_ amount of 1–2 mg despite the presence of hydrophobic folic acid and TCPP. This is a clear pharmaceutical advantage of UCNPs@SiO_2_@FA/PEG/TCPP as a photodynamic agent that should be injected intravenously.

Multimode UV spectrophotometry was used to measure the absorbance profiles between 450 and 750 nm for free TCPP, UCNPs@SiO_2_-NH_2_@FA/PEG, and UCNPs@SiO_2_-NH_2_@FA/PEG/TCPP. As shown in Figure 5C, UCNPs@SiO_2_-NH_2_@FA/PEG/TCPP has a significant absorption peak around 660 nm, a common feature of TCPP [43]. On the contrary, UCNPs@SiO_2_-NH_2_ @FA/PEG without TCPP conjugation had no absorption peak from 450 to 750 nm, showing an almost flat spectrum. The results suggest that UCNPs@SiO_2_-NH_2_@FA/PEG/TCPP would produce a cancer cell death effect by the photodynamic mode.

### 3.4. Singlet Oxygen Generation Assay of TCPP Conjugates

PDT has been considered an effective means of inducing cancer cell death [2,8,65]. To date, several photodynamic therapeutics, such as Photofrin^®^, Ameluz^®^, Metvix^®^, Foscan^®^, Laserphyrin^®^, Visudyne^®^, and Redporfin^®^, have been clinically approved for treating cancers and other indications [66]. Most PDT agents are based on second-generation PS like porphyrin and chlorin, which are more effective, technically superior, and responsive to light of 600–700 nm, with the exception of Visudyne^®^. In our study, TCPP, a derivative of porphyrin, was chosen as a PS surface-modified with UCNPs because it had a good cellular uptake and water solubility [67], whereas many PS show physical aggregation and in vivo delivery problems due to a strong hydrophobicity. By virtue of these properties, TCPP presents an excellent photo-cytotoxicity for treating and visualizing tumors [68] and absorbs ~660–670 nm light emitted from erbium/lutetium-based UCNPs.

Singlet oxygen sensor green (SOSG) is a fluorescent reagent that is highly selective with singlet oxygen and not with hydroxyl radical or superoxide [2,43]. The generated ^1^O_2_ causes SOSG to emit a high green fluorescence (maximum excitation/emission wavelengths = ~504/525 nm) by stopping the quenching process of internal electron transfer. In our study, singlet oxygen generation by respective samples was examined for 60 min. In non-TCPP samples (DW and UCNPs@SiO_2_-NH_2_ @FA/PEG), the SOSG fluorescence level was negligible at a very low value over 60 min (Figure 5D). On the contrary, UCNPs@SiO_2_-NH_2_@FA/PEG/TCPP had almost the same singlet oxygen-generating activity as free TCPP at normalized concentrations of TCPP, and the SOSG fluorescence gradually increased in proportion to the laser irradiation time (Figure 5D). This result indicated that UCNPs@SiO_2_-NH_2_@FA/PEG/TCPP efficiently absorbed 808 nm light and emitted ~660 nm light that activated TCPP to form singlet oxygen. Once again, the efficiency of singlet oxygen generation appeared to be greatly reinforced by the unique ability of the 660 nm-emitting UCNPs platform.

### 3.5. Cytotoxicity Assay for Various UCNPs in Normoxic and Hypoxic Conditions

The cytotoxicities of various UCNPs, including UCNPs@SiO_2_-NH_2_@FA/PEG/TCPP, toward HeLa cells were evaluated through an MTT assay under normoxic and hypoxic conditions. HeLa cells were treated with different concentrations (0, 1, 5, 10, 20, 30 μg/mL) of TCPP-containing samples that were previously normalized as free TCPP, and received 660 and 808 nm laser irradiation for free TCPP and UCNPs, respectively. As shown in Figure 6, in the absence of laser irradiation, all experimental groups showed a low cytotoxicity regardless of the TCPP concentration and oxygen presence. UCNPs@SiO_2_-NH_2_@FA/PEG (without TCPP) displayed an insignificant cytotoxicity either with or without 808 nm light regardless of oxygen due to no lack of singlet oxygen generation. However, UCNPs@SiO_2_-NH_2_@FA/PEG/TCPP appeared to generate singlet oxygen due to a clear cytotoxicity. In fact, UCNPs@SiO_2_-NH_2_@FA/PEG/TCPP killed ~80% of HeLa cells at 30 μg/mL under 808 nm laser irradiation. The overall cytotoxicity degree of UCNPs@SiO_2_-NH_2_@FA/PEG/TCPP plus the 808 nm laser was almost the same as that of free TCPP plus the 660 nm laser, which indicated the effective upconversion of our sample for tumor ablation. Nevertheless, the cytotoxicity of UCNPs@SiO_2_-NH_2_@FA/PEG/TCPP (70% cell viability) was significantly reduced under hypoxic conditions.

### 3.6. Live/Dead Assay for Various UCNPs in Normoxic and Hypoxic Conditions

In live/dead assays, both seeded 2D monolayers and 3D spheroids of HCT 116 cells were treated with PBS, free TCPP, and UCNPs@SiO_2_-NH_2_@FA/PEG/TCPP with or without 808 nm laser irradiation. The overall cytotoxicity results from the Live/Dead^TM^ assay were similar to those of the MTT assay. CLSM images showed the degree of cell death (green emission: live cell, calcein-AM/red emission: dead cell, ethidium homodimer-1). When only PBS was applied, a strong green emission was seen in all samples in normoxic and hypoxic conditions regardless of laser irradiation (Figure 7). In free TCPP, however, almost all cells were killed under a 660 nm laser or normoxic conditions, whereas most cells were live under a no-laser or hypoxic conditions, showing a low intensity of red fluorescence. As with the MTT assay, UCNPs@SiO_2_-NH_2_@FA/PEG had a weak cell killing effect in all cases of laser or oxygen presence. In contrast, UCNPs@SiO_2_-NH_2_@FA/PEG/TCPP showed a strong red emission upon 808 nm laser irradiation, and all HeLa cells appeared to be dead in normoxia. Laser irradiation did not show a significant effect of cancer cell death due to a low production of singlet oxygen in hypoxic conditions. Nonetheless, a significant fraction of cells was dead even in hypoxia for UCNPs@SiO_2_-NH_2_@FA/PEG/TCPP plus the 808 nm laser. As shown in Figure 8A,B, UCNPs@SiO_2_-NH_2_@FA/PEG/TCPP displayed a clear cytotoxicity to cells of ~300 μm-spheroids in normoxia with the 808 nm laser, as proven by the widespread red fluorescence color, and penetrated deep into the spheroids (10–80 μm Z-track images from slices with a 10-μm step size). However, a negligible cytotoxicity was shown in the spheroids treated with UCNPs@SiO_2_-NH_2_@FA/PEG/TCPP in hypoxia with the 808 nm laser and in normoxia without the 808 nm laser (Figure 8B). This finding indicated that UCNPs@SiO_2_-NH_2_@FA/PEG/TCPP would presumably have the practical potential to have a photodynamic antitumor efficacy for the relevant tumor in vivo.

### 3.7. Cellular Uptake of UCNPs@SiO_2_-NH_2_@FA/PEG/TCPP into HeLa Cells

The cellular uptake of UCNP conjugates into HeLa cancer cells was visualized by Bio-TEM and CLSM. As shown in Figure 9A, UCNPs@SiO_2_-NH_2_@FA/PEG/TCPP was internalized into the cells in 10 min (red fluorescence for conjugated TCPP). This result indicates that the UCNPs are taken up by the cells, presumably due to folic acid-mediated endocytosis and small size-based nonspecific uptake [55]. To confirm the internalization of UCNPs@SiO_2_-NH_2_@FA/PEG/TCPP in HeLa cells, cells were photographed by Bio-TEM (Figure 9B). The diameter of HeLa cells was estimated to be 20–40 μm, and UCNPs@SiO_2_-NH_2_@FA/PEG/TCPP was scattered inside or near cells, similar to the original shape of UCNPs@SiO_2_-NH_2_@FA/PEG/TCPP. In addition, the uptake of UCNPs@SiO_2_-NH_2_@FA/PEG/TCPP was observed in an in vitro spheroid model system mimicking tumor tissues in vivo (Figure 9C). The Z-stack images of spheroids were obtained from slices with a 20-μm step size and showed red fluorescence. Based on the spheroid result, UCNPs@SiO_2_-NH_2_@FA/PEG/TCPP was predicted to penetrate into tumors in vivo. Despite the absence of in vivo tumor ablation, we believe that these overall in vitro data demonstrate the great potential for tumor targeting and penetration of our UCNPs platform.

## 4. Conclusions

In summary, highly singlet oxygen-generating upconverting nanoparticles respectively doped with erbium and lutetium for the core and shell (UCNPs@SiO_2_-NH_2_@FA/PEG/TCPP) were fabricated. These nanoparticles were ~50 nm in size and strongly emitted red fluorescence with little green fluorescence, which was considered optimal for the activation of a 660 nm-responsive-photodynamic TCPP agent. The UCNPs generated singlet oxygen and thus killed HeLa cells under 808 nm laser irradiation and normoxic conditions. Furthermore, the UCNPs were well-internalized into the HeLa cell monolayer and three-dimensional cell spheroids, presumably due to the folic acid ligand and small particle size. These overall results suggest that UCNPs could be a potential therapeutic platform for ablating tumors.

## Figures and Tables

**Figure 1 pharmaceutics-12-01102-f001:**
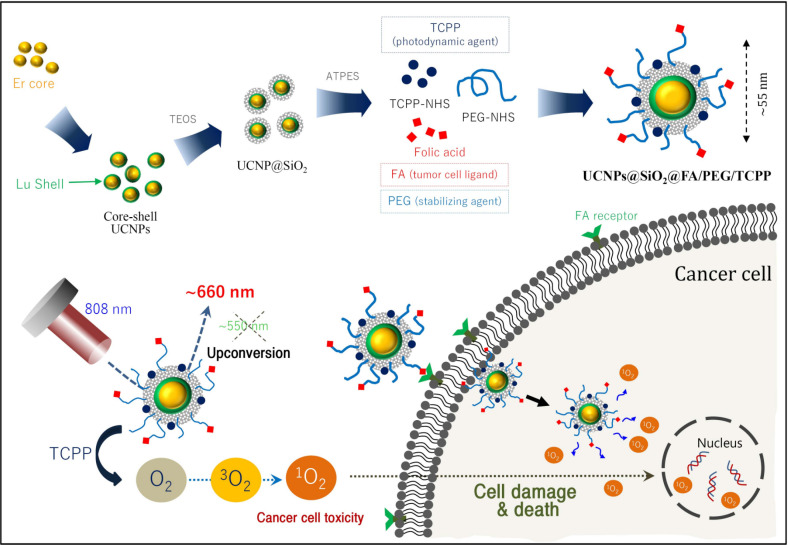
Schematic illustration of the preparation of core-shell upconverting nanoparticles surface-modified with silica, PEG-folic acid (FA/PEG), and TCPP (UCNPs@SiO_2_@FA/PEG/TCPP) for photodynamic therapy based on singlet oxygen generation.

**Figure 2 pharmaceutics-12-01102-f002:**
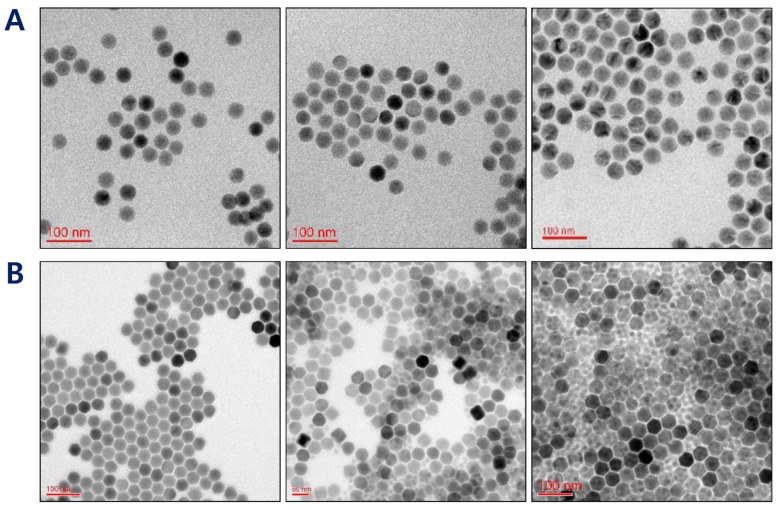
(**A**) TEM images of UCNPs depending upon the ratio between oleic acid (OA) and 1-octadecene (ODE) (3:7 for left, 6:15 for middle, and 4:15 for right). (**B**) TEM images of core@shell UCNPs depending upon the molar ratio between the core (erbium) and shell (lutetium) components (1:0.5 for left, 1:1 for middle, and 1:1.5 for right).

**Figure 3 pharmaceutics-12-01102-f003:**
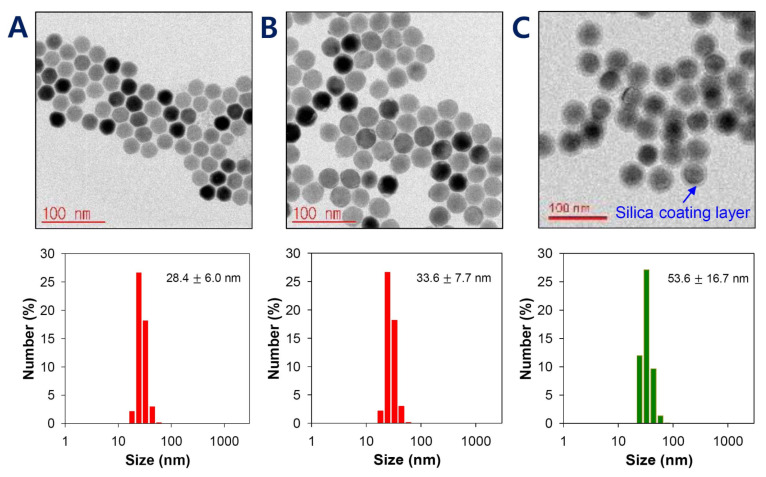
HR-TEM images and histograms of the particle size of (**A**) core-only UCNPs, (**B**) core@shell UCNPs, and (**C**) core@shell UCNPs@SiO_2_. Data are presented as means ± SDs (*n* = 3).

**Figure 4 pharmaceutics-12-01102-f004:**
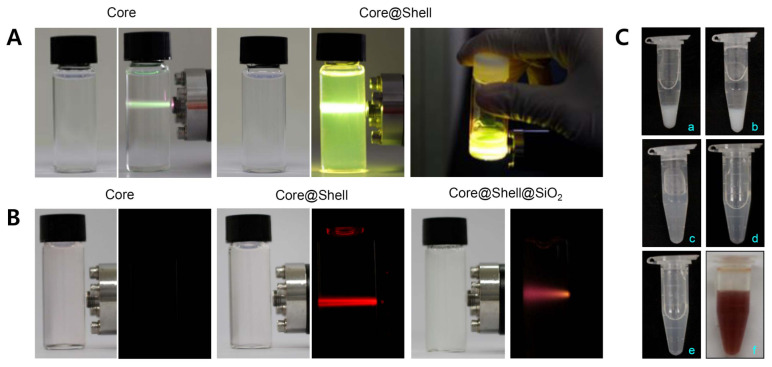
(**A**) Photographs of core or core@shell upconverting nanoparticles (UCNPs) emitting green fluorescence. (**B**) Photographs of core, core@shell, or core@shell@ SiO_2_ UCNPs emitting red fluorescence. (**C**) Photo images of silica-coated upconverting nanoparticles (UCNPs@SiO_2_) (10 mg) modified with (**a**) mPEG (0.1 mg) and FA/PEG (0.1 mg); (**b**) mPEG (0.2 mg) and FA/PEG (0.1 mg); (**c**) mPEG (0.5 mg) and FA/PEG (0.1 mg); (**d**) mPEG (1.0 mg) and FA/PEG (0.1 mg); (**e**) mPEG (2.0 mg) and FA/PEG (0.1 mg); and (**f**) mPEG (2.0 mg), FA/PEG (0.1 mg), and TCPP (1.0 mg).

**Figure 5 pharmaceutics-12-01102-f005:**
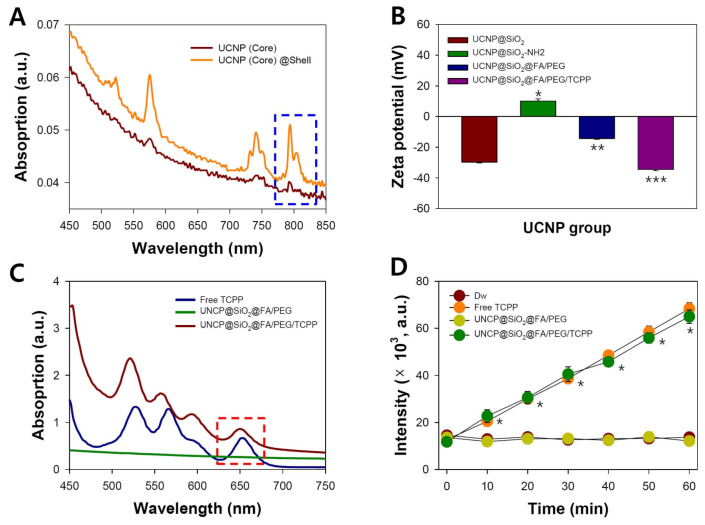
(**A**) UV-Vis-NIR absorption spectra of core and core@shell UCNPs (400–850 nm). (**B**) Zeta potentials of UCNPs@SiO_2_, UCNPs@SiO_2_, UCNPs@SiO_2_@FA/PEG, and UCNPs@SiO_2_@FA/PEG/TCPP (*n* = 3). * *p* < 0.001 over UCNPs@SiO_2_@FA/PEG; ** *p* < 0.005 over UCNPs@SiO_2_@FA/PEG/TCPP; *** *p* < 0.005 over UCNPs@SiO_2_@FA/PEG. (**C**) UV-Vis-NIR absorption spectra (450–750 nm) and (**D**) singlet oxygen generation assay of DW, free TCPP, UCNPs@SiO_2_@ FA/PEG, and UCNPs@SiO_2_@FA/PEG/TCPP (DW: 808 nm, 1.5 W/cm^2^; free TCPP: 0.5 µg/mL, 660 nm, 10 mW/cm^2^; UCNPs@SiO_2_@FA/PEG: 808 nm, 1.5 W/cm^2^; UCNPs@ SiO_2_@FA/PEG/TCPP: 0.35 µg/mL, 808 nm, 1.5 W/cm^2^) (*n* =3). * *p* < 0.001 over Dw and UCNPs@SiO_2_@FA/PEG.

**Figure 6 pharmaceutics-12-01102-f006:**
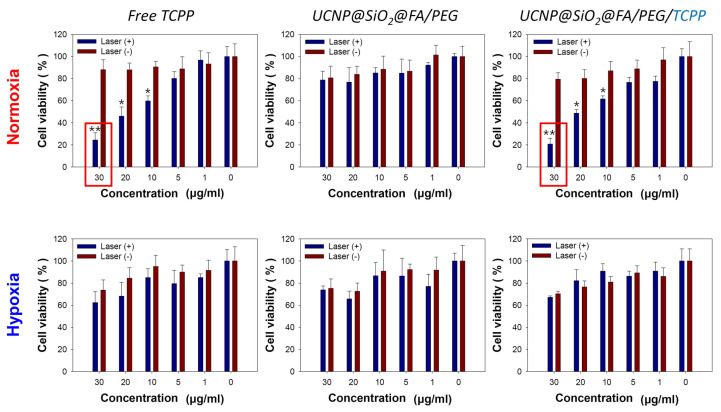
Cytotoxicity evaluation of free TCPP, UCNP@ SiO_2_@FA/PEG, and UCNP@ SiO_2_@FA/PEG/TCPP under normoxic or hypoxic conditions with or without laser irradiation (660 nm, 30 min, 10 mW/cm^2^ for free TCPP; 808 nm, 30 min, 1.5 W/cm^2^ for others). Data are presented as means ± SDs (*n* = 7). * *p* < 0.003 over Laser(−) and ** *p* < 0.001 over Laser(−).

**Figure 7 pharmaceutics-12-01102-f007:**
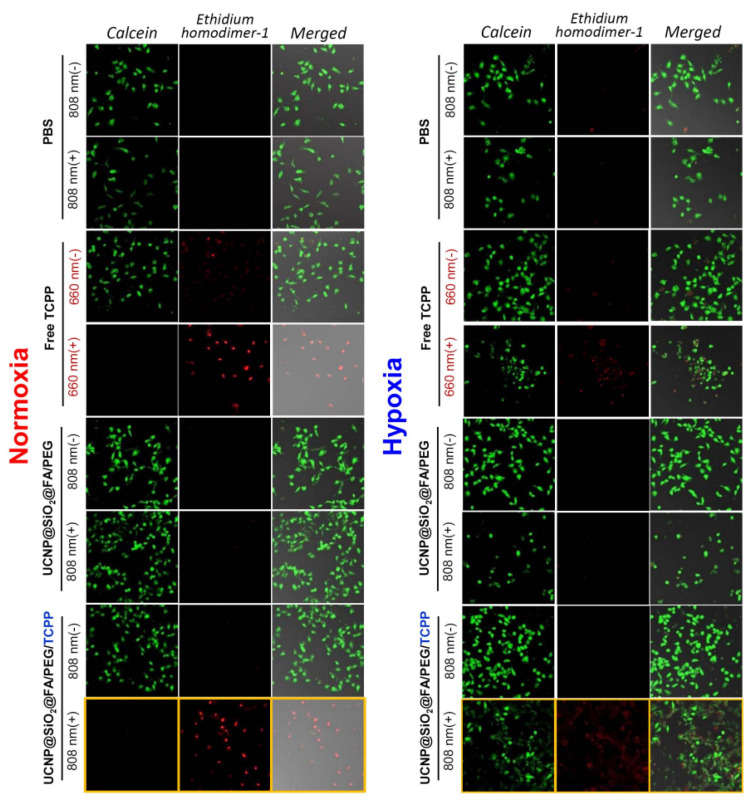
Live/Dead assay of PBS, free TCPP, UCNP@SiO_2_@FA/PEG, and UCNP@SiO_2_@FA/PEG/TCPP under normoxic or hypoxic conditions with or without laser irradiation (660 nm, 30 min, 10 mW/cm^2^ for free TCPP; 808 nm, 30 min, 1.5 W/cm^2^ for others).

**Figure 8 pharmaceutics-12-01102-f008:**
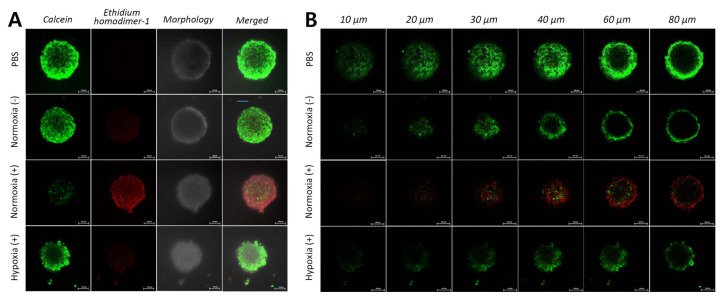
(**A**) Images for the Live/Dead assay in 3D multicellular HeLa cell spheroids under the predetermined conditions (808 nm, 30 min, 1.5 W/cm^2^). (**B**) Z-stack images of six 10–20-μm slices in the HeLa cell spheroids.

**Figure 9 pharmaceutics-12-01102-f009:**
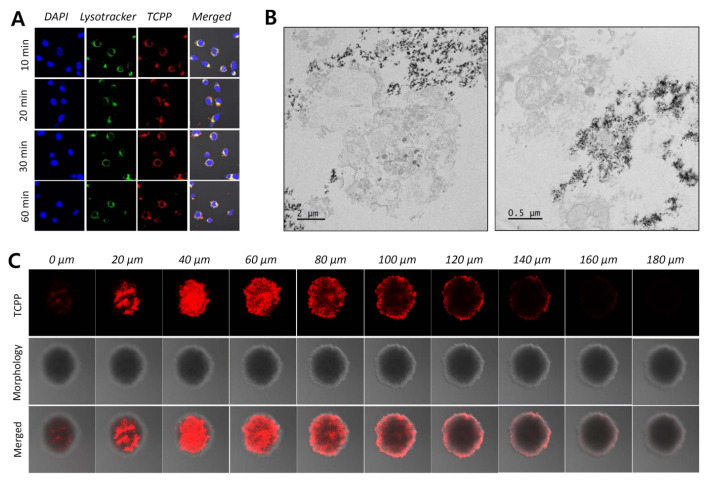
(**A**) Two-dimensional monolayer images of HeLa cells treated with core-shell upconverting nanoparticles surface-modified with silica, PEG-folic acid (FA/PEG), and TCPP (UCNP@SiO_2_@FA/PEG/TCPP) at predetermined time points (blue: nuclei, green: lysosomes, red: TCPP). (**B**) Bio-TEM images of HeLa cells treated with UCNP@SiO_2_@FA/PEG/TCPP. (**C**) In vitro spheroid model images (Z-stack of ten 20-μm slices) for predicting the cellular uptake of UCNP conjugates into three-dimensional tumor tissues.

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
