# Peer review of "Highly Red Light-Emitting Erbium- and Lutetium-Doped Core-Shell Upconverting Nanoparticles Surface-Modified with PEG-Folic Acid/TCPP for Suppressing Cervical Cancer HeLa Cells"

_pharmaceutics, 2020, doi:10.3390/pharmaceutics12111102_

Round 1

Reviewer 1 Report

In this manuscript, several concerns need to be addressed to fit for publication as follows:
General comments:
1. There is a problem with using abbreviations throughout the manuscript. The full term should be mentioned first with the abbreviation between paresis then the abbreviations should be used throughout the manuscript. E.g. line 194: confocal laser scanning microscope (CLSM) then again the full term has been repeated again in line 201,208, and 36. The same situation has been repeated many times throughout the manuscript.
2. The manuscript contains many typing errors, grammatical, formatting, and styling errors. Thus, the manuscript needs to be carefully revised for the English language and formatting by a native English speaker.
Specific comments:
1. Title: photodynamic cancer therapy is a general term. Please, specify the cancer cell line used.
2. Abstract: add " The HeLa cells were 180 treated with each UCNP sample (0, 1, 5, 10, 20, 30 μg/mL as a free TCPP)" to the methods.
3. Introduction: line 60, enumerate the different biomedical applications.
4. Material and methods:
- In the subtitles 2.2., 2.3., and 2.4. , the authors used (UCNPs) for the three different terms. This makes a high confusion.
- Line 178: [44-46] these irrelevant references. Please, use the proper original reference for the method.
- On what basis the tested concentration (0, 1, 5, 10, 20, 30 μg/mL as a free TCPP) has been chosen?
5. Results and discussion:
- In all figure legends, please use the full term, not abbreviations.
- Figures 3, 5, and 6: the authors should symbols denoting the significance and clarify the statistical analysis model used and the number of replicates in the legend.
- Line 232: She shells. revise

Author Response

Response to the 1st reviewer’s comments

In this manuscript, several concerns need to be addressed to fit for publication as follows:
Authors’ Responses: Revision of Ms. ID. pharmaceutics-988670

Highly red light-emitting erbium- and lutetium-doped core-shell upconverting nanoparticles surface-modified with PEG-folic acid/TCPP for photodynamic cancer therapy

Kyungseop Lim, Hwang Kyung Kim, Xuan Thien Le, Nguyen Thi Nguyen, Eun Seong Lee, Kyung Taek Oh, Han-Gon Choi, Yu Seok Youn

General comments:

1. There is a problem with using abbreviations throughout the manuscript. The full term should be mentioned first with the abbreviation between paresis then the abbreviations should be used throughout the manuscript. E.g. line 194: confocal laser scanning microscope (CLSM) then again the full term has been repeated again in line 201,208, and 36. The same situation has been repeated many times throughout the manuscript.

We greatly appreciate the considerate comments for improving the quality of our manuscript. And we are sorry for our technical mistake. We corrected all of the relevant issues according to the reviewer’s suggestions:

(1) Abbreviation and its full name are shown for the first time at each first. CLSM line 211, 220, 227 and 409.
(2) Upconverting nanoparticles and upconversion nanoparticles are unified as upconverting nanoparticles (UCNPs). And upconverting nanoparticles (UCNPs) shown in the title of 2.2., 2.3. and 2.7 were simplified as sole UCNPs.
(3) The full term for TCPP was shown in abstract (because it is kind of independent) first and in 2.2. Materials second. But TCPP in introduction was removed.
(4) Nera-infra red (NIR) is shown in abstract and in introduction twice.
(5) Photodynamic therapy (PDT) is shown in abstract and in introduction twice. And the unnecessary in line 350 was removed.
(6) Singlet oxygen (1O2) is shown in abstract and in introduction twice. But the singlet oxygen word without abbreviation is used elsewhere in the manuscript.

 We authors give many thanks to the reviewer at this considerate comment once again.
2. The manuscript contains many typing errors, grammatical, formatting, and styling errors. Thus, the manuscript needs to be carefully revised for the English language and formatting by a native English speaker.

 We greatly appreciate the considerate comments for improving the quality of our manuscript. We have carefully revised our manuscript according to your suggestion, on the basis of typing errors, grammatical, formatting, and styling errors. Most of all, the current manuscript was revised by an English-correction expert of e-World Editing, one of the most renowned company in Korea. Please find the following certificate of English correction.

Specific comments:
1. Title: photodynamic cancer therapy is a general term. Please, specify the cancer cell line used.

 Yes, the reviewer’s opinion is correct. According to the reviewer’s suggestion, the title was changed as “Highly red light-emitting erbium- and lutetium-doped core-shell upconverting nanoparticles surface-modified with PEG-folic acid/TCPP for suppressing cervical cancer HeLa cells”. Thanks.

2. Abstract: add " The HeLa cells were 180 treated with each UCNP sample (0, 1, 5, 10, 20, 30 μg/mL as a free TCPP)" to the methods.

 We appreciated the careful comment by the reviewer for improving the quality of our article. According to the reviewer’s suggestion, this was added to clarify the study experiment. Thanks.

3. Introduction: line 60, enumerate the different biomedical applications.

 According to the reviewer’s suggestion, this was added to clarify the study experiment. The phrase, “such as, bioimaging, photodynamic therapy and combined cancer therapy” was added. Thank you very much. Please find this at the lines of 62~63.

4. Material and methods:
- In the subtitles 2.2., 2.3., and 2.4. , the authors used (UCNPs) for the three different terms. This makes a high confusion.

  We appreciated the careful comment by the reviewer for improving the quality of our article. As mentioned at #1, the term upconverting nanoparticles and upconversion nanoparticles are unified as upconverting nanoparticles (UCNPs). Furthermore upconverting nanoparticles (UCNPs) shown in the title of 2.2., 2.3. and 2.7 were simplified as sole UCNPs.

- Line 178: [44-46] these irrelevant references. Please, use the proper original reference for the method.

  We appreciated the considerate comment by the reviewer for improving the quality of our article. We’ve just tried to suggest the most similar general method for cytotoxicity (MTT) and live/dead assay performed among our previous papers, not focusing on PDT and Ce6. To be honest, the overall procedures at such a point of view are most well described in these references. As the reviewer indicated, to attain clear citation purpose, we replaced two references among three in the current manuscript. These references perfectly contain all of the contents, MTT assay, live/dead assay, photodynamic therapy, UCNP and HeLa cells. In addition, the phrase “using a slightly modified previously described procedure” was added to avoid misunderstanding in the reference selection.
  And according to the reviewer’s comment, the first two of them were replaced as the references most perfectly matched as follows:

[44] Sun, M,; Xu, L.; Ma, W.; Wu, X.; Kuang, h.; Wang, L.; Xu, C. Hierarchical Plasmonic Nanorods and Upconversion Core–Satellite Nanoassemblies for Multimodal Imaging‐Guided Combination Phototherapy. Adv. Mater. 2016, 28, 898-904.
[45] Yi, Z.; Zeng, S.; Lu, W.; Wang, H., Rao, L.; Liu, H.; Hao, J. Synergistic Dual-Modality in Vivo Upconversion Luminescence/X-ray Imaging and Tracking of Amine-Functionalized NaYbF4:Er Nanoprobes. ACS Appl Mater Interfaces 2014, 6, 3839-3846.

  Please find those. Thank you very much.

- On what basis the tested concentration (0, 1, 5, 10, 20, 30 μg/mL as a free TCPP) has been chosen?

 The TCPP concentration (either free or conjugated) was determined by using an UV-Vis spectrophotometer because TCPP absorbs the 660 nm light without interference of UCNPs or folic acid etc.
 This was added to the method section as follows: “The concentration of conjugated TCPP onto UCNPs was determined at 660 nm using a UV-Vis spectrometer (VersaMaxTM microplate reader, Molecular Devices, CA, USA) because the TCPP absorbance (660 nm) was not overlapped with those of UCNPs alone and folic acid”. Please find this at the lines of 201~204.

5. Results and discussion:
- In all figure legends, please use the full term, not abbreviations.

 We appreciated the careful comment by the reviewer for improving the quality of our article. As the reviewer suggested, we tried to keep the full term of each term in most cases. However, please understand some exceptional cases due to slight difference of each identity that is hard to be added (especially in Figure 5), otherwise the figure legends will be extraordinarily long. According to the reviewer’s suggestion, the abbreviations in figure 1, 4 and 9 was changed. Please find it. Thanks.

- Figures 3, 5, and 6: the authors should symbols denoting the significance and clarify the statistical analysis model used and the number of replicates in the legend.

 We appreciated the careful comment by the reviewer for improving the quality of our article. According to the reviewer’s suggestion, the statistical significances are presented at figure 5B/5D and figure 6. And the number of repetition was clearly added at each legend at figure 3, 5 and 6. Thank you very much.

- Line 232: She shells. Revise

 Thank you very much for this. It is completely corrected: She shell →The shells

Reviewer 2 Report

The manuscript “Highly red light-emitting erbium- and lutetium-doped core-shell upconverting nanoparticles surface-modified with PEG-folic acid/TCPP for photodynamic cancer therapy” is not suitable for publication at the current condition. The manuscript is not organized and presented well. There is a lack of cohesion and coherence. There could also be depth in the discussion. It should be mentioned core-shell whenever TEM/HRTRM supports the statement. For example, the core-shell structure is only shown in Figure 3 (c), except this, there is no other core-shell structure, but the author mentioned core-shell everywhere.

Author Response

Authors’ Responses: Revision of Ms. ID. pharmaceutics-988670

Highly red light-emitting erbium- and lutetium-doped core-shell upconverting nanoparticles surface-modified with PEG-folic acid/TCPP for photodynamic cancer therapy

Kyungseop Lim, Hwang Kyung Kim, Xuan Thien Le, Nguyen Thi Nguyen, Eun Seong Lee, Kyung Taek Oh, Han-Gon Choi, Yu Seok Youn

Response to the 2nd reviewer’s comments

The manuscript “Highly red light-emitting erbium- and lutetium-doped core-shell upconverting nanoparticles surface-modified with PEG-folic acid/TCPP for photodynamic cancer therapy” is not suitable for publication at the current condition. The manuscript is not organized and presented well. There is a lack of cohesion and coherence. There could also be depth in the discussion. It should be mentioned core-shell whenever TEM/HRTRM supports the statement. For example, the core-shell structure is only shown in Figure 3 (c), except this, there is no other core-shell structure, but the author mentioned core-shell everywhere.

Regarding the scientific value of this manuscript, we authors heartily appreciate the considerate comments by the reviewer. As mentioned in the cover letter, this manuscript describes a unique highly red fluorescence-emitting silica-coated core-shell upconverting nanoparticles modified with PEG-folic acid and TCPP as an efficient photodynamic agent for killing tumor cells. Above all, our unique UCNPs exhibited maximum efficiency of emitting red fluorescence light suitable for photodynamic therapy using porphyrin derivatives like TCPP. Also, it displayed pharmaceutical stability based on dispersability in aqueous media and sufficient cancer cell killing activity by generating singlet oxygen.
We believe that that all data set is of high quality and meaningful for the publication, and that our UCNPs@SiO2-NH2@FA/PEG/TCPP would be a new platform of highly efficient and deep-penetrating photodynamic agent for various tumor treatment. We’re assured that this is a first study that includes all these aspects. And please make consideration that recently, photodynamic therapy (PDT) has been considered an effective means of ablating tumors, and upconverting nanoparticles are viewed as an attractive therapeutic platform of anticancer nanoparticles. So, we authors would like to insist that this manuscript would be a timely informative paper.

Regarding manuscript organization and cohesion issue, we’re very happy to have the reviewer’s in-depth comment and suggestion. According to what the reviewer indicated, the manuscript has been revised. Especially, we tried to improve the cohesion text between each paragraph, and this will help the readability by the readers. Please see below. Thank you very much.
(1) Line 243~245
The conventional UCNPs doped with core or shell element of yttrium (Y), ytterbium (Yb) and neodymium (Nd) have split fluorescence lights of ~545 nm and 660 nm, which is considered to be a main reason for reduced PDT efficiency [37]. To this end, the first specific aim of this study was to develop a prototype of UCNPs@SiO2 that emits ~660 nm red light and is surface modified with folic acid for efficient PDT treatment. At a first step, we aimed to produce a hexagonal core of UCNPs to achieve the optimal fluorescence efficacy values [50].
(2) Line 270~273
The size increase at each step of core, shell and silica layer indicated that our UCNPs were doped and coated well with Er, Lu and SiO2, and this unique structure would play a critical role in photodynamic therapy and tumor recognition.
(3) Line 281~283
 The physical stability of UCNPs was a critical consideration in targeting ligand conjugation and further drug delivery because lanthanide-doped and oleate-capped UCNPs are hydrophobic and weakly water-dispersible [51, 52]. Therefore, hydrophilic surface modification of such UCNPs is essential for subsequent pharmaceutical stability and biomedical applications [52].
(4) Line 319~321
 Likewise, core@shell UCNPs showed peak absorption at around 800 nm NIR light, which was much greater than that of core only UCNPs, indicating the importance of the lutetium shell (Figure 5A). The result obviously suggested that these UCNPs would have great effectiveness in generating the relevant light (~660 nm) that is useful for further in vivo PDT, accompanied with the irradiation of deep-penetrating 808 nm NIR laser.
(5) Line 392~293
This result indicated that UCNPs@SiO2-NH2@FA/PEG/TCPP efficiently absorbed 808 nm light and emitted ~660 nm light that activated TCPP to form singlet oxygen. Once again, the efficiency of singlet oxygen generation appeared to be greatly reinforced by the unique ability of 660 nm-emitting UCNPs platform.
(6) Line 436~438
This finding indicated that UCNPs@SiO2-NH2@FA/PEG/TCPP would presumably present practical potential of photodynamic antitumor efficacy to the relevant tumor in vivo.
(7) Line 458~460
Based on the spheroid result, UCNPs@SiO2-NH2@FA/PEG/TCPP was predicted to penetrate into tumors in vivo. Despite the absence of in vivo tumor ablation, we believe that these overall in vitro data demonstrate the great potential of tumor targeting and penetration of our UCNPs platform.

Regarding the use of core@shell term, we authors heartily appreciate the considerate comments by the reviewer. Most of all, the term core/shell is generally accepted in the UCNPs research. Many different lanthanide elements are stacked, mingled, doped and finally structured to form core and shell, independently. Each layer (core and shell) evidently consists in UCNPs, but they are invisible in the TEM images. Nevertheless, this structure is called as a core/shell system that is characterized by the distinct structure layers, being conceptualized in many papers as follows:

In our data, the core and the shell consisted of erbium and lutetium, respectively. And the word “shell” shown in figure 3C is not the one comprising the core@shell UCNPs, but outer silica coating layer. According to the reviewer’s considerate question, in relation with this issue, in order to avoid confusion, the silica “shell” was replaced as just “silica coating layer” in Figure 3C. Please see the figure below.

Round 2

Reviewer 1 Report

The authors adequately responded to all comments and performed the required modifications as directed.

Reviewer 2 Report

The author addressed the comments well and made improvements.